# Geneva Health Forum: The Forum of Innovative Practices in Global Health

**DOI:** 10.3390/ijerph17051517

**Published:** 2020-02-27

**Authors:** Laura Spagnoli, Eric Comte, Danny Sheath, Nicole Rosset, Louis Loutan, Antoine Geissbuhler, Antoine Flahault

**Affiliations:** 1Faculty of Medicine, University of Geneva, 1211 Geneva, Switzerland; eric.comte@unige.ch (E.C.); Danny.Sheath@unige.ch (D.S.); antoine.geissbuhler@unige.ch (A.G.); antoine.flahault@unige.ch (A.F.); 2Division of Tropical and Humanitarian Medicine, Geneva University Hospitals (HUG), 1205 Geneva, Switzerland; Nicole.Rosset@hcuge.ch (N.R.); louis.loutan@hcuge.ch (L.L.)

**Keywords:** global health, access, adapting approaches, technological progress, digital revolution

## Abstract

In the past years, Global Health has interfaced with important challenges related to several dynamic changes. Technological progress, the digital revolution and the emergence of new actors in the field of health, increase the possibility of finding solutions to these unprecedented challenges. Starting from these assumptions, the idea of providing an adequate platform for good management of the health system has flowed into the creation of a meeting place that would allow a wide exchange of information, ideas sharing and proposals for new collaborations: the Geneva Health Forum (GHF). The GHF is a global health conference that aims to promote critical reflections and constructive debates on contemporary global health issues, thus influencing and informing policy formulation with experience from the field. The profile and impact of the Geneva Health Forum has grown year on year, establishing itself as a unique forum, ranging from more traditional sessions to innovative events.

## 1. Introduction: Geneva Health Forum (GHF)—An Innovative Forum

In the past years, Global Health (this expression has emerged to describe the profound shift in the nature of health within the context of globalization [1]), has interfaced with important challenges, reaching fundamental goals for the development of civil society. Among these, we recall the doubling of life expectancies since the end of the 19th century worldwide [2], the eradication of Smallpox in 1980 (which, just in the 20th century, has caused 300 million deaths) [3], the halving in under 5 child mortality in the last three decades [4], and the near elimination of poliomyelitis (which caused paralysis in 350,000 children annually in the 1980s) [5].

Health systems worldwide are facing serious dynamic challenges. It is an age of rapid transformations, ranging from climate change [6,7] to the increase of urbanization [8], from the multiplication of the population to the development of new migration flows [9], and the health needs of populations are equally changeable and demanding, whilst resources become increasingly limited. However, technological progress, the digital revolution, and the emergence of new actors (not strictly medical ones but also, for example digital and social media experts or communication specialists) or new roles of several actors (as the increasingly demanding role of nurses in all settings) in the field of health, increase the possibility of finding solutions to these unprecedented challenges. All decisions made to ensure the effectiveness of health systems should be based on the combination of the best available evidence with an active analysis of the most recent practices. The importance of these objectives is well recognized and accepted, exemplified by the adoption of the Sustainable Development Goals (SDGs) by the United Nations Member States in 2015, which proposed to provide a shared blueprint for peace and prosperity for people and the planet, now and into the future [10,11]. To achieve these objectives, a multidisciplinary approach linked to a continuous exchange of information between all health actors is essential to success.

Starting from these assumptions, the idea of providing an adequate platform for good management of the health system has flowed into the creation of a meeting place that would allow a wide exchange of information, ideas sharing, and proposals for new collaborations: the Geneva Health Forum (GHF).

The GHF is a global health conference that aims to promote critical reflections and constructive debates on contemporary global health issues, thus influencing and informing policy formulation with experience from the field. 

To achieve this goal, past editions of the GHF aimed to give visibility to innovative field experiences but also to establish a critical and constructive dialogue and promote collaborations between global health actors from different sectors, including health practitioners, academics, policy makers, civil society, and the private sector. Another important goal was to link policy and practice, this applies both to guiding policies through best practices and to facilitating policy diffusion at the field level.

At the 2020 edition of the GHF, we explore innovations to improve learning and implementation cycles in global health, to best adapt our practices and improve access to health in a changing world.

## 2. GHF—Previous Editions

The Geneva Health Forum was created in 2006 through a collaboration between the Geneva University Hospitals and the Faculty of Medicine of the University of Geneva, and it is now organized with the collaboration and support of more than 20 partner institutions representing some of the key actors in global health and humanitarian action.

Partners are classified as special partners, which includes the World Health Organization (WHO), Geneva Internationale, The Joint United Nations Programme on HIV/AIDS (UNAIDS), and the United Nations Institute for Training and Research, and other partners (platinum, gold and silver) that are Non-Governmental Organizations (NGOs), foundations, or organizations based in Switzerland. Moreover, in collaboration with the Swiss School of Public Health, the GHF also collaborate with four international global health institutions (American University of Beirut, Harvard Medical School, London School of Hygiene and Tropical Medicine, and the University of Oxford).This collaboration facilitates the meeting of actors in the field of Global Health and the many associations based in Switzerland, particularly in Geneva.

Geneva is a cosmopolitan and health-centered city, where research and scientific development, with particular attention to the health sector, play a fundamental role. It is home to the World Health Organization (WHO) and other UN organizations, permanent missions of 173 countries with their health attachés, major NGOs dedicated to health, including Médecins Sans Frontières (MSF), public and private actors, and a powerful network of academic actors which is often named as the “Health Valley” in Switzerland, beyond Geneva.

The original concept of the GHF was defined around the following four key objectives:Discuss the importance of Universal Health Coverage, exchange experiences, and focus on the best way to improve the access to health care with a particular attention to low-income areas [12].Establish a critical and constructive dialogue between global health actors from different sectors, including academics, policy makers (at national, regional, or international levels), civil society, and the private sector [13].Link the policy with the practice promoting the testimony of actors in the field.Contribute to the dynamism of *La Genève Internationale,* that provides useful links to international organizations, NGOs, and international conferences related to Geneva in the field of health and human rights.

The GHF has multiple formats with the aim of maximizing the opportunities for participants to exchange ideas and engage in discussions about innovative solutions. This particular choice is studied to demonopolize the content from experts to encourage the involvement of all conference participants. The organization of a workshop involves bringing together an expert working group around a specific question over the course of three to four teleconferences during the four months preceding the GHF. The results of the workshops are then presented and discussed during the GHF. At the 2018 edition, twelve workshops involved one hundred and thirty-one participants. These workshops allow people working in similar fields to meet and exchange in depth, not simply presenting a 15 min talk in a parallel session. 

The intention is to make participation of frontline health care workers possible, facilitate their dialogue with policy makers, and help the forging of new and creative alliances. This goal is carried out not only during the three-day forum, but in fact through several meetings that are organized (including recently the GHF Expert Meetings) during the two years between one forum and the next, and interdisciplinary working groups are constantly active, producing articles of high scientific value. An example of the importance of this format was the workshop “Beyond open data: realizing the health benefits of sharing data” at the 2016 edition. All the authors were invited to participate in the discussions, they shared health research data, funded or supported data sharing, or advocated it through their professional position, and this led to publishing an article of high scientific value [14]. Similarly, the next edition the workshop “Leaving no one behind? Reaching the informal sector, poor people and marginalized groups with Social Health Protection” also led to the publication of an article [15].

Each edition is shaped by an open call for abstracts. Following an in-depth review process, these submissions are evaluated and GHF thematic tracks proposed, influenced by the most innovative solutions to the important challenges in Global Health. The selection of the presentations is made carefully, evaluating the proposed topics, the consistency with the theme, and the information that these can provide to all participants. All the information about the thematic, partners, key dates, and the call for abstracts are promoted on the website. This easy way of communication allows organizers to receive 300–400 abstracts; therefore, the review process is strictly controlled and only expert researchers in the global health field are involved in this process.

Each edition gives special attention to the choice of proposed activities, plenary and parallel sessions, workshops, and events that create a stimulating and useful environment for exchange, in particular, the GHF workshops, as described above. 

The first edition took place in 2006 and focused on the state-of-the-art in access to health worldwide. The aim of this theme was to show the different approaches of health systems to help better understand the problems and improve access to health, creating a sustainable global health agenda.

In 2008, the second edition focused on the theme of: “Strengthening of Health Systems and the Global Health Workforce”.

The need to explore this particular theme originated from the evidence showing that Global Health initiatives are important to strengthen any single National Health Systems [16]. This point received global agreement, and several opinions were presented on how to achieve this goal.

The third edition in 2010 was titled “Globalization, Crisis, and Health Systems: Confronting Regional Perspective” and focused on identifying sustainable responses to crises, underlying the importance of local and regional initiatives. In a moment when multiple crises were unfolding, involving finance, food security, and global environmental changes, the debate in GHF 2010 tried to bring the health sector out of isolation, looking not only at strengthening health sector responses but also considering the health consequences and mitigation measures of crises arising from other sectors.

Crises have a double effect on the health system, on the one side, they endanger the accomplishments achieved, but on the other side, they reveal existing weaknesses and disparities, offering the opportunity to formulate potential reforms and innovation.

The important topic of chronic diseases was debated during the fourth edition in 2012: “Growing challenges of chronic diseases”. The growing challenges that Non-Communicable Disease (NCD) pose to health systems in high- and low-income countries was particularly prominent at this edition. The theme of chronic conditions was the result of a consultation process involving hundreds of previous participants and partners.

NCD prevention and management requires a comprehensive and sustainable health system response as well as action across non-health sectors, and that assumption gave the GHF the opportunity to revisit some of its traditional themes, such as health systems strengthening, social determinants of health, and the importance of multi-sectoral action.

In 2014, the fifth edition with the title “Global Health: Interconnected Challenges, Integrated Solutions” took place, covering a wide range of Global Health topics: universal health coverage, health systems, health workforce, innovation, maternal and child health, governance, advocacy, and partnerships.

In this edition, attention was directed to the complexity of global health challenges that involve many sectors and disciplines; therefore, the overarching topic was “integration”.

The GHF invited its participants to revisit the concept of integration in its various dimensions and framings. The results were excellent, with a well-balanced presentation of practical experiences with related projects, programs, and policies.

2016 was marked by significant innovations in terms of governance and program.

The sixth edition, which celebrated the tenth anniversary of the GHF, brought several changes, for example, the reinforcement of the Scientific Committee and Partners with the active participation of new organizations, including the World Health Organization.

One of the most appreciated innovations was the inclusion of a Country Guest of Honor: Germany and France with their World Health Summit [17]. 

The Forum has always been open to all interested people but this particular edition, it was opened to a wider audience (the general public but with particular attention on young researchers), especially thanks to simultaneous translations in English and French for all plenary sessions and at least for two parallel sessions during the whole conference, so language was less of a barrier for many local participants attending this event. 

The theme chosen, “Sustainable and Affordable Innovations in Healthcare”, was equally innovative and gave the opportunity to propose an interactive exhibition space: “Tomorrow’s Affordable Hospital”, which has become one of the characteristic features of the Forum. This demonstration area was a way to collectively present innovation in the same field and to create meaningful connections that could inspire new ideas, new opportunities, and partnerships.

In 2018, the seventh edition, the theme was “Precision Global Health in the Digital Age” and the main objective was to start exploring with this edition the impact of digital tools on public health practices and therefore, examine in which way the digital revolution supports the development of more precise and efficient health interventions [18].

The Russian Federation was the Country Guest of Honor, and the Republic of Tajikistan and the Kyrgyz Republic were the special guests of GHF 2018.

For this edition, the demonstration area included the Global Health Lab, a dynamic and interactive space facilitating exchanges with around 100 different products (Medical devices and Equipment, Pharmaceutical products or Digital Health programs/applications) from several domains of intervention (diagnostic tools, treatment and care devices, training and knowledge management, assistive devices, tools for research, pPatient and community involvement). It was an interactive and dynamic hub where participants were able to try new technologies and products.

## 3. GHF—New Challenges

GHF, faithful to its spirit, will offer in March 2020, the opportunity to have an open and constructive debate on the theme of “Improving access to health: learning from the field”, exploring how innovations can improve learning and implement the learning cycles in global health, to best adapt practices and improve access to health.

The need to explore this issue stems from the belief that to improve the efficiency of learning systems, the health systems should be better organized.

We should study our own practices to learn lessons that will improve the ability to adapt rapidly to the evolution of changing health needs. To achieve these learning systems, researchers, care practitioners, and policy makers need to be better connected. Furthermore, the learning processes must be continuous, so collaborative, multidisciplinary work is essential for success.

New tools, big data management, and innovative technologies also offer opportunities to facilitate such systems, but their use must be carefully considered and closely monitored, ensuring compliance to ethical and legal standards.

To better understand this topic in all its aspects, without forgetting important features, we selected a few questions that we would like to explore as a priority, so we created sub-categories of the main thematic to better address the call for abstracts, helping researchers interested in participating:Improving access to health,Reassessing needs and evaluating impact,Planning and roadmap for action,Implementing change, andLearning global health systems through continuously evaluating impact and adapting approaches.

For the 2020 edition, the Country Guest of Honor is India. This means that India has been involved in the organization of activities and projects, sharing experiences and testimonies of fundamental importance with the public of the GHF.

Furthermore, in order to completely involve all the health actors, two key figures will be included as our special Guests: patients and nurses, inviting the International Alliance of Patients’ Organization (IAPO) and the International Council of Nurses, respectively. The latter will also help mark 2020 as the year of the nurse and midwife.

The presence of patients is a mandatory aspects for us, so, with the help of IAPO, we dedicated several special moments during parallel sessions to better investigate the patients’ point of view. 

The profile and impact of the Geneva Health Forum has grown year on year, establishing itself as a unique forum, ranging from more traditional sessions to innovative events. We are expecting to welcome between 1500 and 1800 participants from approximately 80 countries at the next edition and we aim to increase the active participation of the public, involving it as a main actor and not just a spectator.

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
