# Peer review of "Geneva Health Forum: The Forum of Innovative Practices in Global Health"

_ijerph, 2020, doi:10.3390/ijerph17051517_

Round 1

Reviewer 1 Report

This conference report introduces the Geeva Health Forum (GHF) and describes background, purpose, topics of past meetings as well as topic and expectations of the upcoming meeting to be held in March 2020.

In principle this gives a good overview about the GHF. I have a few comments how this report can be improved further.

In the introduction, please give a brief definition of global health. At the end of the introduction, provide information on what is preesented in the following, i.e. Previous editions and the upcoming meeting. Please give a little more information on the parttner institutions. Are these NGOs, universities etc.? From which countries? Please riefly explain "La Genève Internationale". the authors state that in 2016 the GHF was opened to the general public and young researchers in particular. How was it restricted in the years beforee then? Was it only open to the partner institutions? Please explain the Global Health Lab.

Author Response

We would like to thank editor and reviewers for their careful reading of our manuscript “Geneva Health Forum: The Forum of Innovative Practices in Global Health” and their constructive and useful comments. We believe that the paper is substantially improved due to the reviewers’ inputs.

We have attached a copy of the revised version of the paper in which we have fully addressed, highlighted in yellow, the reviewers’ comments.

Here, you will find the point-by-point answers to all specific comments raised.

Reviewer #1

1) In the introduction, please give a brief definition of global health.
A: Thank you. we added a definition of global health at the beginning of the introduction.

2) At the end of the introduction, provide information on what is presented in the following, i.e. Previous editions and the upcoming meeting.
A: Thank you. We have reviewed this and we added more information about next edition of the GHF: at the next edition of the GHF, we will explore innovations to improve learning and implementation cycles in global health, to best adapt our practices and improve access to health in a changing world.

3) Please give a little more information on the partner institutions. Are these NGOs, universities etc.? From which countries?
A: We better specified our partners information: Partners are classified as special partners, which includes the World Health Organization, Geneva Internationale, UNAIDS and the United Nations Institute for Training and Research, and other partners (platinum, gold and silver) that are Non-Governmental Organizations (NGOs), Foundations or Organizations based in Switzerland. Moreover, in collaboration with the Swiss School of Public Health the GHF collaborate also with four International Global Health institutions (American University of Beirut, Harvard Medical School, London School of Hygiene & Tropical Medicine and University of Oxford).

4) Please briefly explain "La Genève Internationale".
A: we included a brief description about La Geneve Internationale: that provides useful links to international organizations, NGOs and international conferences related to Geneva in the field of health and human rights.

5) The authors state that in 2016 the GHF was opened to the general public and young researchers in particular. How was it restricted in the years before then? Was it only open to the partner institutions?
A: We clarified this point stating that The Forum has always been open to all interested people but this particular edition it was opened to a wider audience (the general public but with particular attention on young researchers) especially thanks to simultaneous translations in English and French for all plenary sessions and at least for two parallel sessions during the whole conference; so language was less of a barrier for many local participants attending this event.

6) Please explain the Global Health Lab.
A: We better explain what is the Global Health Lab stating that it's a demonstration area, a dynamic and interactive space facilitating exchanges with around 100 different products (Medical devices and Equipment, Pharmaceutical products or Digital Health programmes/ applications) from several domains of intervention (Diagnostic Tools, Treatment and care devices, Training and Knowledge Management, Assistive devices, Tools for Research, Patient and community involvement). 

Reviewer 2 Report

Thank you for the opportunity to read this report. I appreciated the historical summary of the evolution of the GHF. A longer article on this history might be of interest at some point.

I have a few comments for your consideration. These are meant to help explain the GHF to the general reader, who may not have attended a GHF, or who may be interested in submitting a proposal.

  • in the Intro section, who/what are the "new actors" or "new roles" referred to?
  • what is the governance structure of the Geneva Health Forum? Who organizes it? Who constitutes its peer review committee (for abstracts, for example)? Where does the call for abstracts appear (where is it advertised)?
  • who funds it?
  • can you briefly give a few specific or concrete examples of papers or contributors to illustrate the range of presenters? For instance on p. 2 you refer to the importance of front line participation -- here an example of past front line participation would help to illustrate the GHF's record on this point.
  • how do participants get involved in the working groups between forums?
  • in the section GHF - new challenges, I don't know what this sentence means: "sub-categories have been designed to better address the call for abstracts." Pls clarify this.
  • the drive to include patients is laudatory, as is the openness to the public. How will you achieve these goals?What does success look like for the GHF in this regard?

Author Response

We would like to thank editor and reviewers for their careful reading of our manuscript “Geneva Health Forum: The Forum of Innovative Practices in Global Health” and their constructive and useful comments. We believe that the paper is substantially improved due to the reviewers’ inputs.

We have attached a copy of the revised version of the paper in which we have fully addressed, highlighted in yellow, the reviewers’ comments.

Here, you will find the point-by-point answers to all specific comments raised.

1) Thank you for the opportunity to read this report. I appreciated the historical summary of the evolution of the GHF. A longer article on this history might be of interest at some point.I have a few comments for your consideration. These are meant to help explain the GHF to the general reader, who may not have attended a GHF, or who may be interested in submitting a proposal.
A: Thank you very much for this useful comment.

2) in the Intro section, who/what are the "new actors" or "new roles" referred to?
A. as suggested we clarified the differences between new roles and new actores stating that technological progress, the digital revolution and the emergence of new actors (not strictly medical ones but also, for example digital and social media experts or communication specialists) or new roles of several actors (as the increasingly demanding role of nurses in all settings) in the field of health, increase the possibility of finding solutions to these unprecedented challenges.

3) what is the governance structure of the Geneva Health Forum? Who organizes it? Who constitutes its peer review committee (for abstracts, for example)? Where does the call for abstracts appear (where is it advertised)?
A: Thanks for this comment, we already specified that The Geneva Health Forum GHF was created through a collaboration between the Geneva University Hospitals (HUG) and the Faculty of Medicine of the University of Geneva (UniGe), and that now it's organized with the collaboration of more than 20 partner institutions. We have clarified which partners support the organization the GHF and we added information about the call for abstract "All the information about the thematic, partners, key dates and the call for abstracts are promoted on the website; this easy way of communication allows organizers to receive 300-400 abstracts, therefore the review process is strictly controlled and only expert researchers in the global health field are involved in this process."

4) who funds it?
A: The key funders of the GHF are University Hospital of Geneva (HUG) and University of Geneva (UniGe), also the Swiss Agency for Development and Cooperation (SDC). In addition, all of our partners make a small financial contribution to the running of the forum.

5) can you briefly give a few specific or concrete examples of papers or contributors to illustrate the range of presenters? For instance on p. 2 you refer to the importance of front line participation -- here an example of past front line participation would help to illustrate the GHF's record on this point.
A: Thanks for this point. As suggested, we added two concrete examples of workshops that resulted in the publication of two articles of high scientific value.

6) how do participants get involved in the working groups between forums?
A: As suggested we clarified how workshops are organized stating that the organization of a workshop involves bringing together an expert working group around a specific question over the course of three to four teleconferences during the four months preceding the GHF. The results of the workshops are then presented and discussed during the GHF. At the 2018 edition, twelve workshops involved one hundred and thirty one participants.These workshops allow people working in similar fields to meet and exchange in depth, not simply presenting a 15 minute talk in a parallel session.

7) in the section GHF - new challenges, I don't know what this sentence means: "sub-categories have been designed to better address the call for abstracts." Pls clarify this.
A: As suggested, we have reviewed the section GHF - new challenges and we have added the sentence: "To better understand this topic in all its aspects, without forgetting important features, we selected few questions that we would like to explore as a priority, so we created sub-categories of the main thematic to better address the call for abstracts, helping researchers interested in participating".

8) the drive to include patients is laudatory, as is the openness to the public. How will you achieve these goals?What does success look like for the GHF in this regard?
A: The presence of patients is a mandatory aspects for us, so, with the help of International Alliance of Patients' Organization (IAPO), we dedicated several special moments during parallel sessions to better investigate the patients' point of view.